# Analysis of the Snake Robot Kinematics with Virtual Reality Visualisation

**DOI:** 10.3390/s23063262

**Published:** 2023-03-20

**Authors:** Anna Sibilska-Mroziewicz, Ayesha Hameed, Jakub Możaryn, Andrzej Ordys, Krzysztof Sibilski

**Affiliations:** 1Institute of Micromechanics and Photonics, Department of Mechatronics, Warsaw University of Technology, 02-525 Warsaw, Poland; 2Institute of Automatic Control and Robotics, Department of Mechatronics, Warsaw University of Technology, 02-525 Warsaw, Poland; 3Air Force Institute of Technology, 01-494 Warsaw, Poland

**Keywords:** Virtual Reality, snake robot, simulation of multi-body systems, kinematics

## Abstract

In this article, we present a novel approach to performing engineering simulation in an interactive environment. A synesthetic design approach is employed, which enables the user to gather information about the system’s behaviour more holistically, at the same time as facilitating interaction with the simulated system. The system considered in this work is a snake robot moving on a flat surface. The dynamic simulation of the robot’s movement is realised in dedicated engineering software, whereas this software exchanges information with the 3D visualisation software and a Virtual Reality (VR) headset. Several simulation scenarios have been presented, comparing the proposed method with standard ways for visualising the robot’s motion, such as 2D plots and 3D animations on a computer screen. This illustrates how, in the engineering context, this more immersive experience, allowing the viewer to observe the simulation results and modify the simulation parameters within the VR environment, can facilitate the analysis and design of systems.

## 1. Introduction

Biomimetic robots imitate living organisms’ appearance, shape, or behaviour and are designed to utilise biological principles to replicate natural behaviour and solve complex problems. A snake robot is an example of a biomimetic robot characterised by its high level of redundancy and numerous degrees of freedom. The robot’s movement is produced by changes in its internal shape, similar to the motion of natural snakes. The robot’s joints, which link its segments, are defined by a series of angles that describe its configuration [1].

Virtual Reality (VR) has an important role in Industry 4.0, the fourth industrial revolution, and offers new ways to use VR in robotics applications through virtual manufacturing. Integrating VR with robotics has several potential uses [2], including enabling the planning of robot trajectories through trial and error in VR rather than using existing industrial systems; training operators; assisting surgeons in robotic surgery procedures; developing safe procedures for human-robot collaboration; and creating an integrated environment for control design that encompasses models (digital twins), control algorithms, and VR. VR simulation provides a more immersive and engaging way to study simulation results. In the virtual environment, abstract properties of the motion, such as angles or velocity vectors, can be displayed. The VR application also allows the user to change their perspective, stop or rewind the simulation, and manipulate simulation parameters within the VR environment.

VR provides a cost-effective approach to controlling virtual robots in a simulated environment and facilitates the training of operators. VR is also a valuable tool for testing complex simulations involving human–robot collaboration. In [3], the authors developed a virtual environment for validating mathematical models of robots. In addition, multiple users can be tracked in VR while performing assembly tasks, thereby aiding in designing effective workplace layouts for humans and robots in the industrial setting [4].

Robot-assisted surgery is a promising field that employs VR simulators to train doctors in various clinical procedures, including those using da Vinci robots. The immersive perspective of VR visualisation provides medical students with an unprecedented understanding of anatomy, enabling the exploration of organs at both the micro and macro scales. Moreover, immersive, dynamic models of physiological and pathological processes could result in an experience of “immersive medicine” [5].

Programming by Demonstration is a technique for teaching robotic systems new behaviours from a demonstration by a human operator. To reduce programming complexity, robots can be taught new movements in VR environments [6]. A novel methodology based on heuristic beam search, described in [7,8], has been implemented in VR. This algorithm plans collision-free paths for *n* degree-of-freedom robots, with human involvement in defining the free space or collision-free volume and selecting the start and goal configurations.

To ensure safe and efficient collaboration between humans and robots, planning the workspace layout and programming of the industrial robotic work cell is essential. VR can aid in achieving these tasks. Ref. [7] highlights the use of VR in planning the workspace layout and programming the robotic cell, which can improve the safety of human–robot collaboration. Furthermore, VR-enhanced Computer-Aided Design (CAD) software provides an effective way to create and visualise an appropriate layout for the robotic cell.

VR has been demonstrated to enhance learning for high school students by simplifying and simulating complex concepts across various fields. Virtual Reality experiments have also been found to improve student understanding in science courses and to increase their interest in learning through immersive experiences [9,10,11,12]. Virtual Reality laboratories provide a safe environment for students to perform experiments without the risk associated with real-world materials or hazardous situations. In mathematical subjects, geometric models can be taught effectively using VR, as students can easily visualise complex geometry models to improve the quality of their education.

One further application of Virtual Reality is testing and implementing control algorithms for various systems. In [13], a solution is proposed for decentralised formation tracking of groups of mobile robots using consensus and Model Predictive Control. The solution is designed to ensure collision avoidance and communication topology, with experimental results verified in a VR environment. In [14], an illusion-based VR interaction technique is proposed where the virtual hand of a user is moved during a reach to guide their physical hand to a specific location. The proposed method is validated for developing a control approach to achieve redirection and desired point and path tracking. Additionally, VR simulations are used to test and train autonomous transition control methods for drones, which assist farm workers in scouting to improve the efficiency of vineyard management [15], and in smart cities [16].

VR is a technology that allows the visualisation and interaction with three-dimensional environments, including objects’ behaviour. It is used to simulate and study complex environments for entertainment and research purposes. VR interfaces are used for visualisation, interaction with robotics, planning, usability, and infrastructure for both experts and non-experts [17]. However, the use of VR with biomimetic robotics has not been extensively researched. More specifically, there is limited research on using snake robots in VR. However, the motion of haptic snakes has been investigated for multiple feedback methods, such as tapping and gesture feedback [18].

A comprehensive review is presented in the literature from 2015 to 2019 that covers VR applications in medical robotics [19]. In [20], the author presented the VR challenges for manufacturing systems for industry 4.0, which conclude that the adoption of immersive technologies of AR/VR systems in manufacturing industries has persistently increased. Moreover, research possibilities are in areas that improve the flexibility of multiple users in a VR environment and the development of methodologies for simultaneous interface and concurrent interactions among collaborators in a VR environment. Another promising area is a mixed reality framework for human–robot collaboration [21].

The dynamical model of the snake robot and its kinematics was described in several articles [22,23]. In this article, it was shown that when the snake robot moves on the ground, the friction or drag force coefficients of the snake robot are larger in a sideway direction than in the longitudinal direction of the link. However, there is a lack of visual implementation. Previous articles on the subject analyse the joint angles’ angular velocities and the robot’s head or mass centre linear velocity [24,25,26,27,28]. In these articles, snake robots used Line-of-Sight (LOS) guidance control law to exponentially stabilise the desired straight-line path under a given condition on the look-ahead distance parameter. However, our article describes Point-of-Sight (PoS) control law. Moreover, our article presents a new method for enhancing kinematic studies by analyzing the velocity in all segments’ normal and tangential directions. Simulation results include plots generated in MATLAB and screenshots taken in a VR environment.

In this paper, we present a new approach for evaluating control systems in snake robots. Currently, Virtual Reality (VR) and augmented reality (AR) are employed to visualise the motion of machines and multi-body systems, utilising virtual entities and presenting information in standard graphs and numerical data. In particular, AR offers additional information often imperceptible in the physical world. In this study, we introduce supplementary layers of information, such as vectors (e.g., velocity, friction, torque) and colours, to facilitate a more comprehensive understanding of how parameter changes impact the control system’s quality in snake robots. We employ a synesthetic design approach to enable insightful evaluations of the algorithms used in robotics. This paper proposes a new method for presenting engineering simulation results in VR. Specifically, it investigates the movement of a snake robot on a flat surface and scrutinises how various model parameters affect its motion. The study utilises MATLAB to compute the robot’s dynamic model and control algorithms while the Unity engine generates the virtual environment and animations.

This article is structured into five sections. Section 1 introduces the use of VR technology in engineering and advances in biomimetic snake robots. Section 2 discusses the dynamic model of the snake robot, the control algorithm used for the robot to reach the destination position, and the implementation details of the applications created in MATLAB and Unity. The simulation results are presented in Section 3. The obtained results are discussed in Section 4. Finally, Section 5 provides the conclusions and future work.

## 2. Materials and Methods

### 2.1. Snake Robot Model

The model of the snake robot depicted in the article is based on widely used equations of snake robot motion on a flat horizontal surface. A detailed description of the model can be found in [22,29]. This dynamical model can be derived based on the torque equilibrium equation:(1)$$M_{\theta }\ddot{\theta }+W{\dot{\theta }}^{2}-lS_{\theta }Kf_{R,x}+lC_{\theta }Kf_{R,y}=D^{T}u$$
where: $S_{\theta }=diag\left(sin\left(\theta \right)\right)\in R^{N\times N}$ and $C_{\theta }=diag\left(cos\left(\theta \right)\right)\in R^{N\times N}$ are square matrices with trigonometric functions of link angles at the diagonal and zeros in the remaining elements, and link angles $\theta ={\left[\theta_{1},\ldots ,\theta_{N}\right]}^{T}\in R^{N}$ in a global coordinate system.

To define the robot’s configuration, there is a differentiation between link angles $\theta_{i}$ and joint angle $\phi_{i}$ indicated in Figure 1. A link angle is defined as an angle between the link and a global *x*-axis, while a joint angle is a difference between the link angles of two neighbouring links $\phi_{i}=\theta_{i}-\theta_{i+1}$. The vector $u\in R^{N-1}$ defines the controllable parameters–actuator torques exerted on successive links. The $f_{R,x}$ and $f_{R,y}$ vectors represent components of friction force on the links in global *x*- and *y*-directions. This model assumes viscous friction forces acting on the mass centre of the links. The friction force is defined in a local coordinate system attached to each robot’s segments, as shown in Figure 1. Friction force is proportional to the segment’s linear velocity $v_{i}$ and coefficients $c_{t}$ and $c_{n}$:(2)$$f_{R,i}=-\left[\begin{array}{cc} c_{t} & 0\\ 0 & c_{n} \end{array}\right]v_{i}$$

Anisotropic friction force enables the snake robot’s movement by producing lower friction coefficients, denoted as $c_{t}$, in the joints’ longitudinal direction compared to the normal direction, where the coefficient is denoted as $c_{n}$. This difference in coefficients allows the joints to slide forward.

### 2.2. Path Following Controller

A simple approach for path-following is the Line-of-Sight (LoS) method [30], which involves moving towards a series of pre-defined reference points. This method requires the robot to follow a straight line between its current and target positions. Once the robot reaches the desired accuracy for the current target position, it moves on to the next reference point in the sequence.

The gait pattern of a snake robot’s commonly used control system is lateral undulation, described in [31]. The dynamical analysis of various snake robot motion patterns can be found in [32], and the performance of different motion strategies is discussed in [33]. For the lateral undulation pattern, each joint angle *i* of the robot, where *i* belongs to the set 1,…,N−1, is controlled using the following equation
(3)$$\phi_{i,\mathrm{ref}}=\alpha \sin\left(\omega t+\left(i-1\right)\delta \right)+\phi_{o}$$
where α and ω are the amplitude and angular frequency, respectively, of the sinusoidal joint motion, δ determines the phase shift between the joints, and $\phi_{0}$ is a joint offset, which we assume to be identical for all joints. The joint offset controls the direction of the locomotion and allows the robot to reach the destination point. It is defined as the difference between the heading angle $\bar{\theta }$ and heading reference angle ${\bar{\theta }}_{ref}$ as
(4)$$\phi_{o}=k_{\theta }\left(\bar{\theta }-{\bar{\theta }}_{\mathrm{ref}\;}\right)$$

The controller gain $k_{\theta }>0$ influences the control system efficiency. The heading angle of the snake robot is defined as an average of link angles:(5)$$\bar{\theta }=\frac{1}{N}\sum_{i=1}^{N}\theta_{i}$$

The heading reference angle ${\bar{\theta }}_{ref}$ designates a direction to the reference position as follows
(6)$${\bar{\theta }}_{ref}=-\arctan\left(\frac{p_{y}}{p_{x}}\right)$$
where $p_{x}$ and $p_{y}$ are distances between the robot’s head and destination point along the *x*- and *y*-axes of the inertial coordinate system. The joint torque is determined according to the control law given as
(7)$$\bar{u}=k_{p}\left(\phi_{\mathrm{ref}}-\phi \right)-k_{d}\dot{\phi }$$

The system performance depends on controller gains $k_{p}>0$ and $k_{d}>0$.

### 2.3. MATLAB Simulations

The equations of motion for the snake robot were coded in MATLAB software and solved using the ode solver. Additionally, control algorithms were incorporated into the software implementation, enabling the snake robot to reach a specified position while keeping track of its centre. This program can simulate the system for various parameter sets, including adjustments to the target position, friction coefficients, and controller gain.

Two coordinates define the destination position in the inertial coordinate system. A series of points can also be defined, and once the robot reaches one of them, it will move on to the next in the sequence. If the algorithm does not define the subsequent target point, the robot continues to move forward using the last calculated value of the joint offset.

The friction coefficients in the normal and tangential directions are the primary parameters determining the snake robot’s behaviour. A distinct difference in the anisotropy of friction force is necessary for the robot to move forward. The simulation program lets the user define the viscous friction coefficient within 0.1 to 10 (Ns/m).

To direct a robot towards a target orientation, we manipulate a joint offset defined in Equation (Equation 4). The control algorithm’s effectiveness is determined by the controller gain $k_{\theta }>0$, which can vary between 0 (when the robot’s dynamics are not influenced by path following) and 3.

The simulation generates several plots, such as the robot’s trajectory, joint angles, and control signals, as well as graphs depicting the heading angle and reference heading angle’s progression. Additionally, each segment’s resultant position and orientation at discrete simulation intervals are saved to a txt file or transmitted directly to the visualisation program. This data allows visualisation programs to accurately reproduce the snake robot’s motion.

The MATLAB implementation is available at GitHub webpage (https://github.com/asibilska/Snake-Robot-Locomotion-MATLAB-, Retrieved 3 February 2023) and Matlab Central webpage (Snake-Robot-Locomotion-MATLAB, MATLAB Central File Exchange. https://uk.mathworks.com/matlabcentral/fileexchange/102910-snake-robot-locomotion-matlab Retrieved 3 February 2023). A full description of the program and implemented snake robot model can be found in [34].

### 2.4. Visualisation of the Snake Robot Motion

#### 2.4.1. Simulink 3D Animation

Initially, the robot’s movement was visualised using Simulink 3D Animation Toolbox, a MATLAB library that utilises the Virtual Reality Markup Language (VRML). The segment geometry was imported from the STL file created in SOLIDWORKS. MATLAB calculated the segment positions and orientations, which were saved in a txt file to create the animation. Figure 2 shows that this visualisation provided a deeper understanding of the snake robot’s motion from various perspectives. However, manipulating the camera in the program was challenging, and the graphics of the solution were relatively inadequate. Thus, a different visualisation approach was employed for the snake robot.

#### 2.4.2. Three-Dimensional Simulations in Unity

The latest version of the 3D visualisation for the snake robot was developed using Unity, a popular game engine known for its advanced graphical animations. Besides creating three-dimensional and two-dimensional games, Unity is also used in several industries, including film, automotive, architecture, engineering, and construction. Its intuitive editor has drag-and-drop functions and scripting abilities based on the widely used C# language. The Unity engine offers a comprehensive training platform with numerous tutorials, examples, and specialised training paths, making it an ideal choice for this application.

To improve VR software development, a useful tool is the Software Development Kit (SDK), which offers a collection of pre-built and configured interactions for VR projects. The XR Interaction Toolkit, which is one of the most popular free SDKs, is used in our project. Other widely used solutions include the Oculus Interaction SDK and the Windows Mixed Reality Toolkit. The SDK provides script libraries to implement interactions in VR projects, such as grabbing objects, interacting with the user interface, recognising hand gestures, locomotion systems, and physics interactions.

The implemented program allows running simulations on Oculus Quest 2 (Meta Quest documentation, http://developer.oculus.com/documentation, Access date: 3 February 2023), a VR headset developed by Oculus, a division of Meta. It is a standalone device that can run games and software wireless under an Android-based operating system. It supports positional tracking with six degrees of freedom, using internal sensors and an array of cameras in the front of the headset rather than external sensors. An Oculus Quest 2 set consists of VR goggles with a resolution of 3664 × 1920 (1832 × 1920 per eye) and a refresh rate of 90 Hz, and two symmetric controllers enabling tracking of the user’s hands. They allow the user to interact with the virtual environment by pointing at objects with a ray or pressing the controller’s buttons.

The visualisation program utilises the CAD-designed geometry of the snake robot segments. Using data generated by MATLAB, the program interpolates the position and orientation of the segments in consecutive moments. The Unity scene, as shown in Figure 3, features an immersive environment, lighting, and a camera that tracks the user’s head movement. Additionally, a Graphical User Interface (GUI) allows the user to interact with certain elements of the scene. The user can move around in either continuous or teleportation mode. The GUI contains grabbable and interactable parts, such as the reference position or buttons.

There is also an optionally available grid that features white marks that are 1(m) apart and a red mark that indicates the origin of the Cartesian coordinate system.

One of the main benefits of visualising the snake robot’s movement in VR is the user’s ability to observe it from any distance or perspective. This means the user can change the camera position or move around the virtual scene. To switch to a different pre-defined camera position, the user must press one of the controller buttons, X/Y/A/B, as shown in Figure 4.

There are five sets of camera positions available, each offering a unique perspective:No button pressed—default stationary camera at eye-level position and horizontal orientation. This camera observes the GUI in front of the user, and the snake robot is seen from a height like it is moving on the floor.The B button pressed—the camera is facing downwards perpendicular to the floor. It moves with a robot’s centre and rotates to track the $\bar{\theta }$ angle.The A button pressed—this camera simulates a camera attached to the robot’s head. It is moving and rotating along with the first segment of the robot.The Y button pressed—the camera is stationary and is facing downwards, perpendicular to the floor.The X button pressed—the camera is in a low position behind the target position. It changes position when the robot reaches subsequent targets.

The user can manipulate the Oculus Quest 2 controllers to control the snake robot’s simulation. The user can increase or decrease the simulation’s speed, pause, or rewind it. The trigger button shown in Figure 5 can be used to stop or rewind the simulation, with the amount of pressure applied to the button determining the magnitude of the rewind. The user can decelerate the simulation by pressing a push button, and the deceleration rate depends on the force applied. Further, haptic feedback (controller vibrations) may occur when reaching a designated position.

The Unity game engine provides a way to display an interactable Graphical User Interface (GUI) in VR. The GUI consists of elements such as sliders, drop-down lists, and push/toggle buttons, and the user interacts with them by casting a ray from the controllers, as shown in Figure 6. In our program, the GUI is stationary and located in front of the default camera position. It enables the user to restart the simulation and change the data source. There are two simulation modes available. In the first “off-line” mode, an application reads previously calculated data from a text file generated by MATLAB. In this mode, the GUI displays a drop-down list with all files uploaded to the application.

In the second “real-time” mode, the robot’s position and orientation are calculated in real-time while the visualisation runs on Unity. In this case, the simulation in MATLAB and VR visualisation run in parallel. Unity and MATLAB communicate via TCP Sockets to exchange information about simulation parameters and trajectories. In “real-time” mode, users can assign the robot’s destination point and change the simulation parameters in GUI, such as friction coefficients in the normal and tangential directions and the path following gain. The GUI provides sliders to change parameter values in a given range. Application modes can be changed by marking the “data form file/data from MATLAB” checkbox in GUI.

The VR GUI consists of checkboxes that allow users to turn on/off the visualisation of selected physical properties of the robot’s motion, such as the segment’s velocities and guidance angles. The segment’s velocity is depicted as a vector attached to the segment’s centre. The velocity of the segment is calculated as the quotient of the change in its position by the time of data sampling. The user can select by GUI slider which segment’s velocity should be displayed.

There is a panel shown in Figure 7 attached to the user’s left hand. It shows the current values of selected parameters:time,heading angle,reference heading angle,position of the snake robot’s head in global X−Y coordinates.

Plots are generated in MATLAB, and display:X−Y position of robot head and remaining segments,heading angle versus reference heading angle,joint angles,link angles,tangential and normal components of segment velocities,absolute and global X−Y components of the snake robot’s head velocity.

The application also provides visualisation of the guidance angles. The heading reference angle, denoted by ${\bar{\theta }}_{\mathrm{ref}}$, is displayed as a red line that connects the destination point, the robot’s head, and the horizontal line. Meanwhile, the heading angle, denoted by $\bar{\theta }$, is represented by a yellow line.

The VR environment includes a red cylinder to represent the location of the destination point that the robot is following. In “off-line” mode, the destination position and reaching time are saved in a file. In “real-time” mode, the user marks the desired position on the floor with controller rays and clicks the trigger button to confirm the new position. The application sends the reference point to MATLAB via a TCP protocol, and the control algorithm calculates the target trajectory. The robot’s segment’s calculated positions are sent back to Unity, and the robot’s configuration is displayed in VR. If the user assigns a new target position, the algorithm restarts, and the snake robot returns to the origin. This allows the user to observe the algorithm’s performance and compare results for different parameters.

## 3. Results

For kinematic analysis of the snake robot, described by mathematical model Equation 1, we have carried out the following numerical studies for varying parameters:friction coefficients in normal $c_{n}$ and tangential $c_{t}$ directions;parameters of the lateral undulation gait pattern, Equation (Equation 3): α, ω, δ;controller gains, $k_{p}$ and $k_{d}$, described in control law Equation (Equation 7) and $k_{\theta }$ described in joint offset Equation (Equation 4).

For all analysis, we have assumed the same robot’s geometrical parameters:number of robot segments: N=10;the segment mass: m=1;the segment length: l=0.2;the moment of inertia: $J=ml^{2}/3$.

### 3.1. Friction Coefficients

The anisotropy of the friction coefficient was introduced in Equation (Equation 2), where $c_{t}$ is the friction coefficient in the tangential direction and $c_{n}$ in the normal direction. During studies, we have analysed four cases:$c_{t}=0.1$ and $c_{n}=10$;$c_{t}=1$ and $c_{n}=10$;$c_{t}=10$ and $c_{n}=10$;$c_{t}=10$ and $c_{n}=1$.

All simulations in this subsection were performed for:(8)$$\phi_{i,\mathrm{ref}}=40^{\circ }\sin\left(40^{\circ }t+\left(i-1\right)43.5^{\circ }\right)$$
and
(9)$$\bar{u}=1\left(\phi_{\mathrm{ref}}-\phi \right)-2\dot{\phi }$$

#### 3.1.1. Robot’s Configurations for Different Friction Coefficients

Figure 8, Figure 9, Figure 10 and Figure 11 display different sets of parameters and their effects on the robot’s configurations. Each figure contains three screenshots taken from the VR application, with one plot per figure.

The MATLAB-generated plot displays the positions of the robot’s head (indicated by a thick blue line) and its remaining segments (indicated by dotted lines) on the *x*- and *y*-axes. The robot’s configuration at t=5 seconds, t=10 seconds, and t=24 seconds is marked on the plot with pentagrams, circles, and triangles, respectively. Moreover, the bottom plot shows the position of the robot’s head along the *x*-axis, while the left plot shows its position along the *y*-axis.

At the 5th, 10th, and 24th seconds of the simulation, screenshots were taken in VR. The camera used in these shots was fixed and placed parallel to the floor, as seen in Figure 4, where the camera is attached to the Y button. A grid representing the coordinate system was displayed in the virtual scene.

Figure 10 indicates that the robot requires anisotropy of friction coefficients to move. When $c_{t}=c_{n}$, the robot seems to slide on the surface without any forward motion. However, as shown in Figure 8 and Figure 9, when $c_{t}<c_{n}$, the robot slides forward. The displacement is greater when there is a larger difference between coefficients. On the other hand, as depicted in Figure 11, when $c_{t}>c_{n}$, the snake moves backwards, and the head’s trajectory remains a polygonal chain rather than a sinusoidal curve.

#### 3.1.2. Analysis of Linear Velocities of Robots Segments

Figure 12 displays the velocity of the snake robot’s head. The plot contains three lines: the yellow line represents the absolute velocity value, the red line corresponds to the global *y*-component of velocity, and the blue line shows the *x*-component of velocity. At three points in time, t=6(s), t=9(s), and t=11(s), there are markers on the plot indicating the velocity values.

The oscillations along the *y*-axis in the global coordinate system have a similar appearance in the first three simulations, with an amplitude of approximately A=0.2(m) and a period of about T=5(s). However, the amplitude of oscillation is considerably lower in the fourth simulation. The oscillations along the *z*-axis in the global coordinate system have a mean value of zero in the third simulation, resulting in no forward movement of the robot. In the first and second simulations, the mean value of the *x*-axis velocity is positive, causing the robot to move forward. The resultant velocity is significantly higher in the first case. Finally, in the fourth simulation, the *x*-axis velocity oscillates around a negative value when the robot moves backwards.

Figure 13, Figure 14, Figure 15 and Figure 16 depict the velocities of each robot segment, with the yellow line indicating the velocity magnitude. The blue line represents the tangential component of velocity, which is aligned with the *x*-axis of the local coordinate system of each segment, and the red line represents the normal component of velocity, which is aligned with the *y*-axis of the local coordinate system of each segment.

Figure 12, Figure 13, Figure 14 and Figure 15 provide insights into how the robot’s segments move, whether they slide longitudinally along the segment’s length or laterally, perpendicular to the segment. In the first two simulations (Figure 12 and Figure 13), segments 1, 2, 9, and 10 primarily move tangentially, while segments 4, 5, and 6 predominantly move in the normal direction. All segments exhibit dominant normal velocity directions in the third and fourth simulations.

The simulation results are also presented in Figure 17, Figure 18, Figure 19 and Figure 20. The top-left plot displays the absolute velocity value of each segment (top plot) and the ratio of the tangential velocity to the absolute velocity value (bottom plot). The VR screenshots illustrate the robot’s configuration, with arrows representing the velocity vectors of each segment at various simulation time points.

The results obtained support the previous findings. As seen in Figure 17 and Figure 18, the arrows are primarily oriented tangentially for segments 1, 2, 9, and 10 and perpendicularly for segments 4, 5, and 6. In contrast, the arrows in Figure 19 and Figure 20 are predominantly normal for all segments.

### 3.2. Parameters of Gait Pattern

The motion of the snake robot is characterised by a lateral undulation pattern defined by Equation (Equation 3) and determined by the amplitude α, angular frequency ω, and phase shift δ. The subsequent section examines the impact of each of these parameters on the robot’s motion. In all simulations, we used the values $c_{t}=1$, $c_{n}=10$, $k_{p}=1$, and $k_{d}=2$.

#### 3.2.1. Amplitude

We conducted a series of simulations with an angular frequency of $\omega =40{(}^{\circ }/s)$, a phase shift of $\delta =60^{\circ }$, and a sequence of amplitudes: $\alpha_{1}=30^{\circ }$, $\alpha_{2}=40^{\circ }$, $\alpha_{3}=50^{\circ }$, and $\alpha_{4}=60^{\circ }$. Figure 21, Figure 22, Figure 23 and Figure 24 display the robot trajectories, which indicate that increasing the amplitude results in a more pronounced bending of the robot’s body, and a significantly greater distance travelled by the robot, as observed in the VR screenshots.

Figure 25 and Figure 26 depict plots of the link angles θ (top-left), joint angles ϕ (top-right), link angle angular velocities $\dot{\theta }$ (bottom-left), and joint angle angular velocities $\dot{\phi }$ (bottom-right). The thick line represents the mean angle value. The larger α values correspond to larger link and joint angles. The derivatives of the angles also increase with larger amplitudes.

Figure 26 shows the velocity of the robot head. The absolute value of the velocity is marked by the yellow line, the component of the velocity along the *x*-axis of the global coordinate system is marked by the blue line and the component of the velocity along the *y*-axis of the global coordinate system is marked by the red line. The robot’s velocity increased with larger α; however, there is no significant difference between $\alpha =50^{\circ }$ and $\alpha =60^{\circ }$.

Figure 27 shows the torques applied to the robot’s joints. They increased linearly with rising amplitude.

The simulation results have been gathered in Table 1. Column *d* indicates the distance travelled by the robot’s head after 25 seconds of simulation; max(|u|) is the maximum value of torque applied to the robot’s joints; *V* is the final velocity of the robot head in the 25th second of the simulation; max(|θ|), max(|ϕ|) are the maximum value of the link and joint angle; finally, $max(|\dot{\theta }|)$ and $max(|\dot{\phi }|)$ are the maximum values of angle derivatives. While calculating columns 5, 7–10, we considered only the system’s steady state after the 200th simulation sample.

#### 3.2.2. Angular Frequency

The next series of simulations investigated the influence of angular frequency ω on the snake motion. The simulations were performed for α=50(deg/s), $\delta =60^{\circ }$, and sequence of frequencies $\omega_{1}=30{(}^{\circ }/s)$, $\omega_{2}=40{(}^{\circ }/s)$, $\omega_{3}=50{(}^{\circ }/s)$, $\omega_{4}=60{(}^{\circ }/s)$. Figure 23, Figure 28, Figure 29 and Figure 30 indicate the change in the robot’s shape. For smaller ω, the robot’s body is more curved and bent. Figure 31 indicates changes in robot angles and their derivatives. For larger frequencies, the values of angles are decreasing. However, their derivatives are increasing. Nevertheless, the difference in derivatives is not so significant as for different amplitudes, shown in Figure 25. There is no significant influence of frequencies on the snake head’s velocity, as shown in Figure 32. An increase in angular frequency strongly influences the values of torques in joints, as shown in Figure 33. The simulation results for different frequencies have been gathered in Table 2.

#### 3.2.3. Phase Shift

Analogous experiments were performed for $\alpha =50^{\circ }$, $\omega =50{(}^{\circ }/s)$, and a sequence of different phase shifts: $\delta_{1}=30^{\circ }$, $\delta_{2}=40^{\circ }$, $\delta_{3}=50^{\circ }$, $\delta_{4}=60^{\circ }$. As we can see from Figure 29, Figure 34, Figure 35 and Figure 36, the robot’s shape is the same but the movement direction changes. The values of joint angles, derivatives of joint angles, and torque remain the same for all phase shift values (see Figure 37, Figure 38 and Figure 39, and Table 3).

### 3.3. Controller Gains

This section presents the results of simulations for different controller gains. The simulation results have been collected in Table 4. All simulation were performed for $\alpha =50^{\circ }$, $\omega =50^{\circ }$, $\delta =60^{\circ }$, $c_{t}=1$, and $c_{n}=10$.

The distance travelled by the robot increases with the $k_{p}$ coefficient. The relation between distance and $k_{d}$ does not look so obvious. For $k_{p}=1$ and $k_{d}=0.5$ the robot behaves chaotically. The required torque seems to increase with rising kp and lower with $k_{d}$. There is no unambiguous trend between gain coefficients and joint/link angles.

### 3.4. Path Following Coefficient

The last parameter considered in our studies was $k_{\theta }$, which allows the robot to follow a destination point. For these simulations, we have assumed $\alpha =50^{\circ }$, $\omega =50^{\circ }$, $\delta =60^{\circ }$, $c_{t}=1$, $c_{n}=10$, $k_{p}=3$, and $k_{d}=3$, and a destination point situated in coordinates (5,5). The simulations were performed for $k_{\theta }=0.1$, $k_{\theta }=0.3$, $k_{\theta }=0.5$, $k_{\theta }=1$, $k_{\theta }=2$, and $k_{\theta }=3$. Larger gains increase the distance travelled by the robot and improve the tracking of the reference heading angle, as shown in Figure 40 and Figure 41. However, this means the significant increase in control signals shown in Figure 42 and joint angles shown in Figure 43 can violate design constraints. The $k_{\theta }$ above the value of one causes unacceptable torque values and can destroy the robot’s mechanism. The optimal solution is to assign a path-following coefficient larger than $k_{\theta }=0.5$ to ensure path-following but smaller than $k_{\theta }=1$.

## 4. Discussion

Our research aimed to introduce a novel method of studying robotic multi-body systems utilising Virtual Reality technology. By integrating VR visualisation in Unity and simulation in MATLAB, we could conduct a series of real-time, interactive simulations of the motion of a snake robot. We could adjust the robot’s parameters, designate destination points using VR controllers and GUI, and observe an animation illustrating the robot’s behaviour. The VR environment allowed us to follow the motion from multiple perspectives, pause or rewind the simulation, analyse the velocities and heading angles of different segments, and compare the animations with the plots produced in MATLAB.

Our research generated a collection of experiments demonstrating how various parameters affect the robot’s motion. The necessary condition for the movement of a snake robot is the anisotropy of frictional force. Conducted studies show that the robot moves faster with a greater difference in friction in the tangential and normal directions. Increasing the amplitude of the gait pattern causes the robot to move faster, but it may lead to a violation of joint constraints. The angular frequency and phase shift determines the robot’s direction. Choosing optimal controller gains, $k_{p}$ and $k_{d}$, is a difficult task that requires further investigation for different scenarios. According to our studies, the optimal value of the path following coefficient $k_{\theta }$ is between 0.5 and 1.

In the research, we utilised the snake robot as a representative example of a multi-body dynamical system featuring non-linearity. Our study demonstrates that even minor alterations to the parameters or controller gains can result in significant variations in the robot’s performance. Identifying the optimal set of parameters to maximise the robot’s performance under diverse environmental conditions and destination positions is challenging. Hence, our research emphasises the implementation of advanced control algorithms, such as Model Predictive Control or Sliding Mode Control, to operate the snake robot. We will conduct VR-supported studies to test and evaluate the efficacy of these control algorithms.

## 5. Conclusions

Virtual Reality is a cutting-edge technology. It allows users to experience and interact with virtual environments. It has applications in entertainment, education, and socialisation and has the potential to be a game-changer in engineering research. In addition, VR has immense potential to revolutionise research in the engineering field. A pioneering application of VR in visualising snake robot motion is presented in this article. This application could significantly influence the development of control algorithms for robotic systems.

At the moment, we are in the process of developing sophisticated control algorithms for a snake robot. This will enhance the VR application by visualising the abstract control properties and extending its functionality.

To this end, VR, Unity, and Matlab integration can be utilised to conduct realistic simulations and design the digital twin for mechanical systems. Therefore, several potential future research proposals have been identified based on the findings of this study. First, this approach will enable a synesthetic approach to enhance the control system design methodology. Exploring the best ways to present information to improve the engineering understanding of control systems would be worthwhile. Building on the current research, novel visual evaluation methods for mechanical control systems based on augmented reality can be developed.

In the specific case of evaluating snake robot controllers, it is possible to use VR to visualise additional physical parameters of the robot during motion, such as friction forces, joint torques, and link angle errors. Furthermore, ongoing investigations evaluate using the snake robot’s Model Predictive Control (MPC) in various control scenarios involving obstacles or joint failures. Ghost robots can be displayed to indicate the predicted positions of the robot.

## Figures and Tables

**Figure 1 sensors-23-03262-f001:**
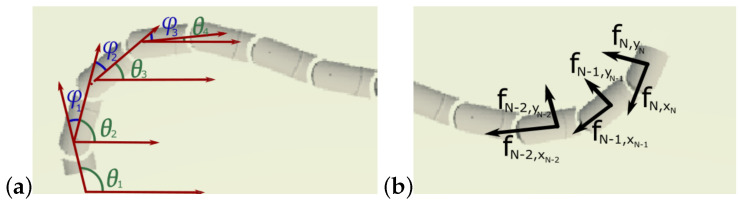
Assignment of (**a**) joint and link angles (**b**) friction force in the normal and tangential direction.

**Figure 2 sensors-23-03262-f002:**
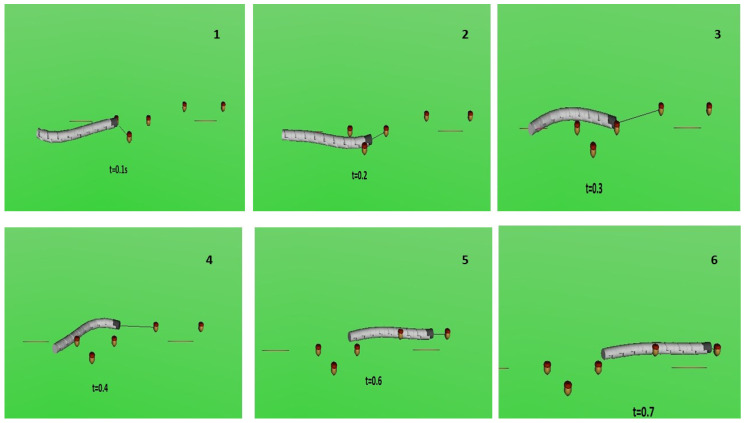
Visualisation of the snake robot implemented in Simulink 3D Animation.

**Figure 3 sensors-23-03262-f003:**
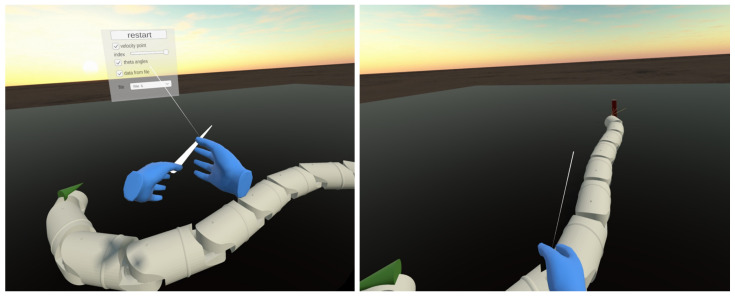
Virtual scene with the snake robot implemented in Unity.

**Figure 4 sensors-23-03262-f004:**
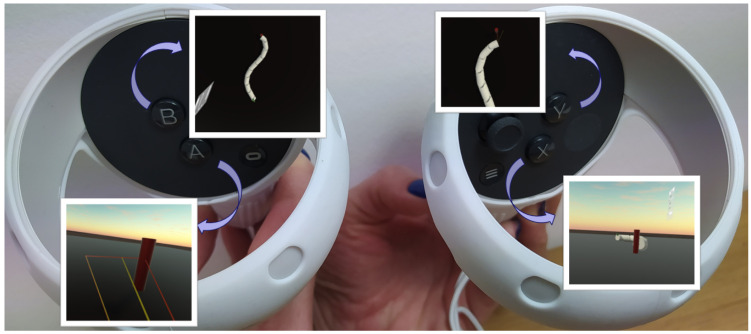
Oculus Quest 2 controllers with buttons changing the camera position.

**Figure 5 sensors-23-03262-f005:**
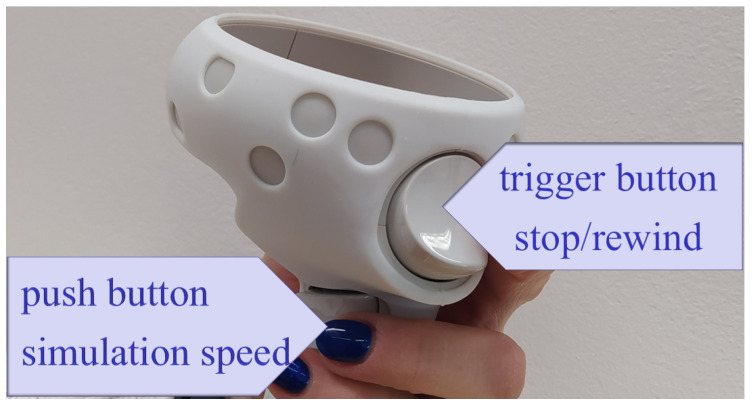
Oculus Quest 2 controllers with buttons used to control the simulation speed.

**Figure 6 sensors-23-03262-f006:**
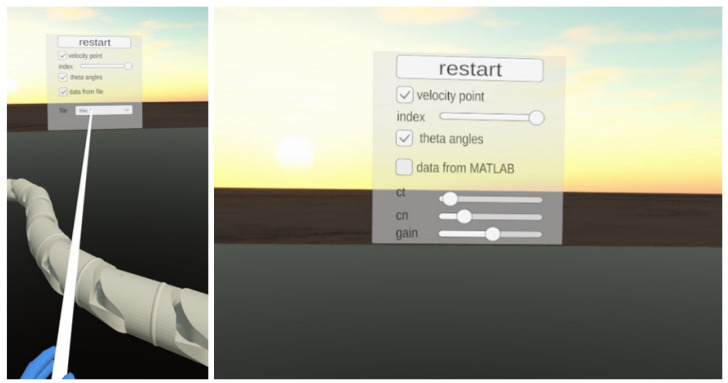
Graphical User Interface and controllers ray allowing interaction with GUI.

**Figure 7 sensors-23-03262-f007:**
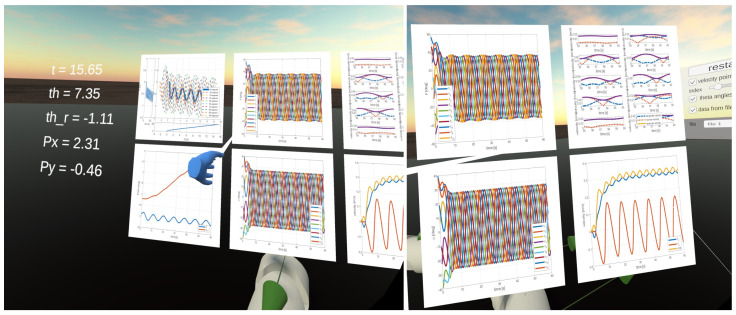
Screen from VR application showing a panel with simulation parameters and plots generated in MATLAB.

**Figure 8 sensors-23-03262-f008:**
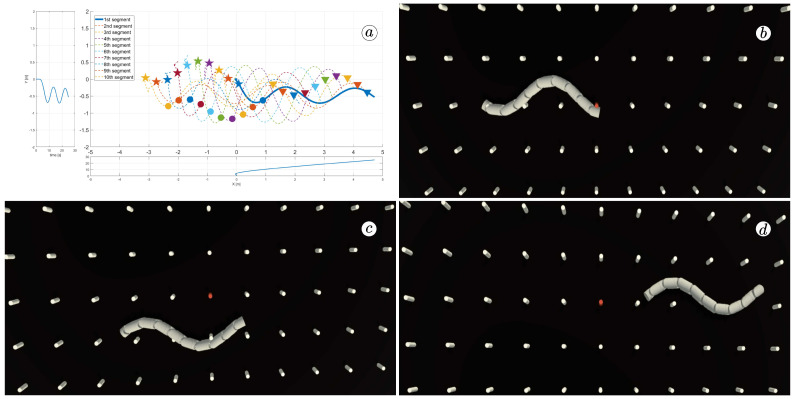
Trajectories of robot segments for $c_{t}=0.1$ and $c_{n}=10$ (**a**) and robot configurations for t=5(s) (**b**), t=10(s) (**c**), and t=24(s) (**d**).

**Figure 9 sensors-23-03262-f009:**
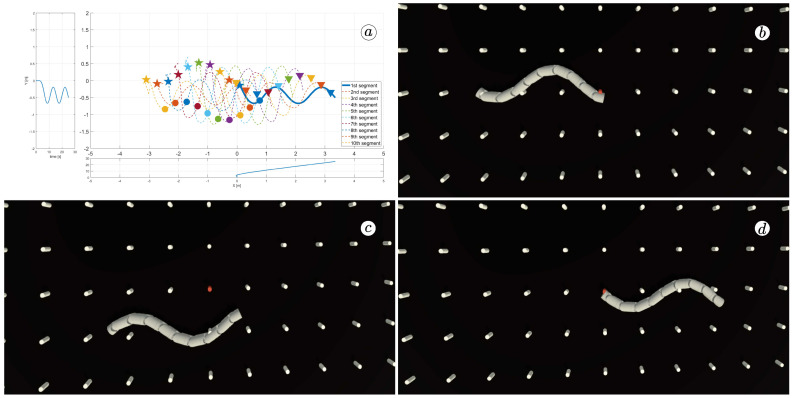
Trajectories of robot segments for $c_{t}=1$ and $c_{n}=10$ (**a**) and robot configurations for t=5(s) (**b**), t=10(s) (**c**), and t=24(s) (**d**).

**Figure 10 sensors-23-03262-f010:**
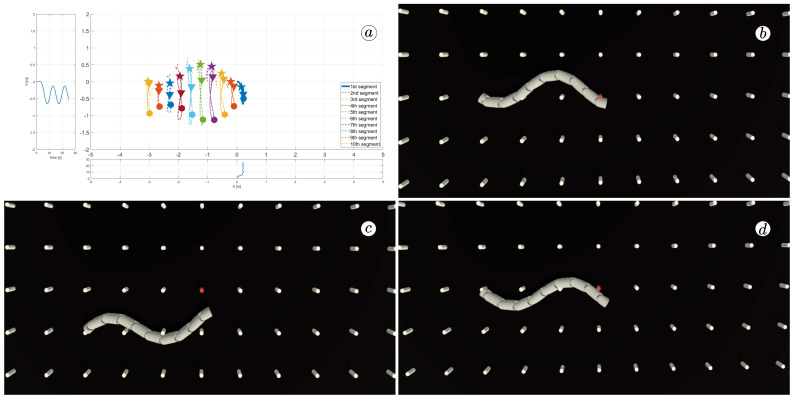
Trajectories of robot segments for $c_{t}=10$ and $c_{n}=10$ (**a**) and robot configurations for t=5(s) (**b**), t=10(s) (**c**), and t=24(s) (**d**).

**Figure 11 sensors-23-03262-f011:**
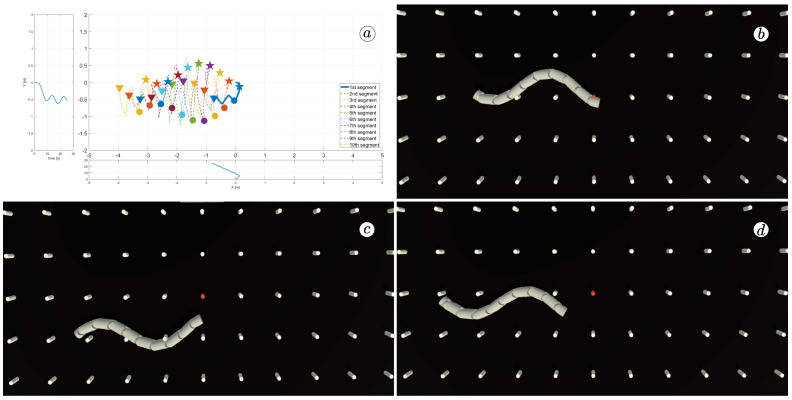
Trajectories of robot segments for $c_{t}=10$ and $c_{n}=1$ (**a**) and robot configurations for t=5(s) (**b**), t=10(s) (**c**), and t=24(s) (**d**).

**Figure 12 sensors-23-03262-f012:**
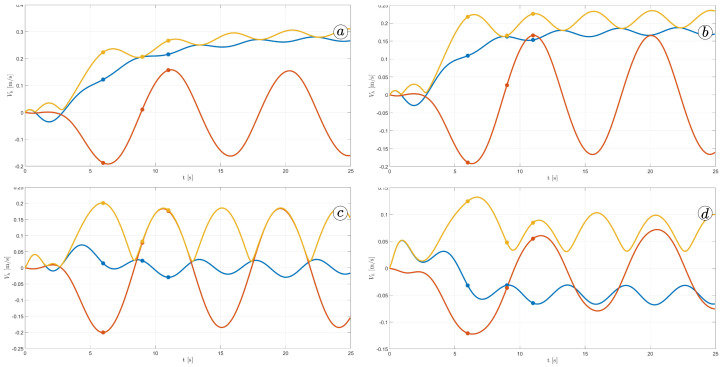
Velocity of the snake robot’s head for $c_{t}=0.1$ and $c_{n}=10$ (**a**) $c_{t}=1$ and $c_{n}=10$ (**b**), $c_{t}=10$ and $c_{n}=10$ (**c**), and $c_{t}=10$ and $c_{n}=1$ (**d**).

**Figure 13 sensors-23-03262-f013:**
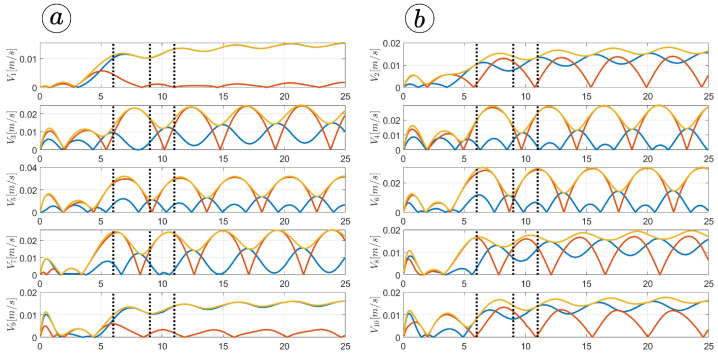
Velocities of the snake robot’s segments for $c_{t}=0.1$ (**a**) and $c_{n}=10$ (**b**).

**Figure 14 sensors-23-03262-f014:**
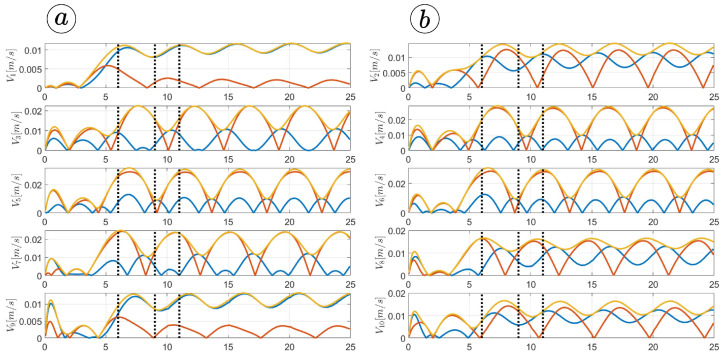
Velocities of the snake robot’s segments for $c_{t}=1$ (**a**) and $c_{n}=10$ (**b**).

**Figure 15 sensors-23-03262-f015:**
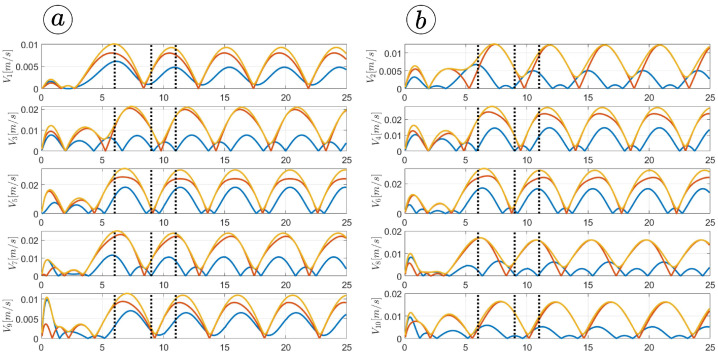
Velocities of the snake robot’s segments for $c_{t}=10$ (**a**) and $c_{n}=10$ (**b**).

**Figure 16 sensors-23-03262-f016:**
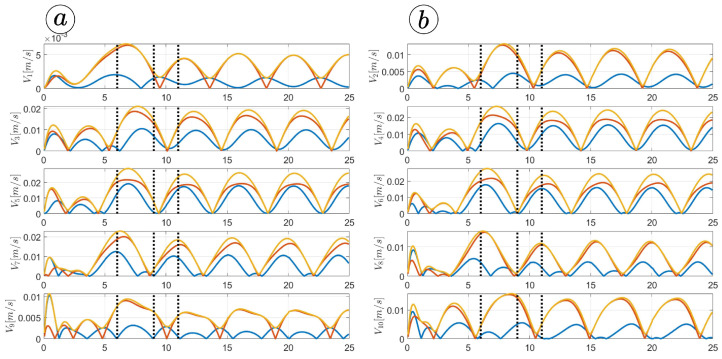
Velocities of the snake robot’s segments for $c_{t}=10$ (**a**) and $c_{n}=0$ (**b**).

**Figure 17 sensors-23-03262-f017:**
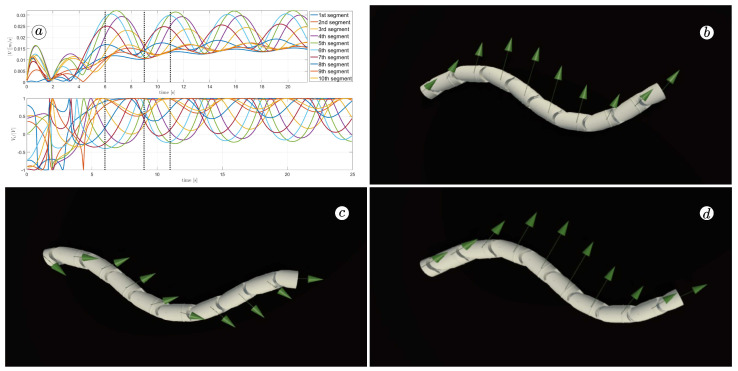
Absolute value and tangential component of segments velocities for $c_{t}=0.1$ and $c_{n}=10$ (**a**) and velocity vectors for t=6(s) (**b**), t=9(s) (**c**), and t=11(s) (**d**).

**Figure 18 sensors-23-03262-f018:**
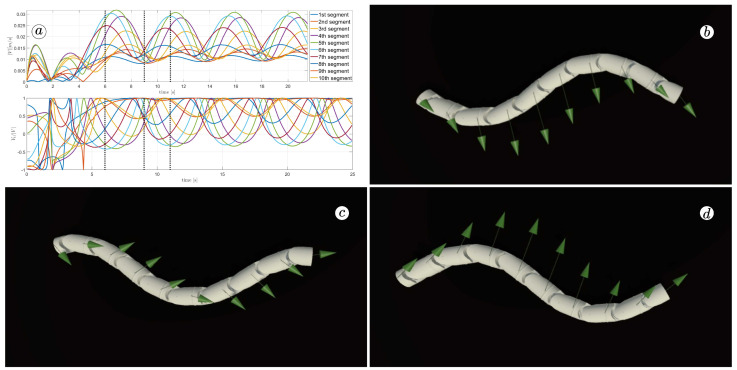
Absolute value and tangential component of segments velocities for $c_{t}=1$ and $c_{n}=10$ (**a**) and velocity vectors for t=6(s) (**b**), t=9(s) (**c**), and t=11(s) (**d**).

**Figure 19 sensors-23-03262-f019:**
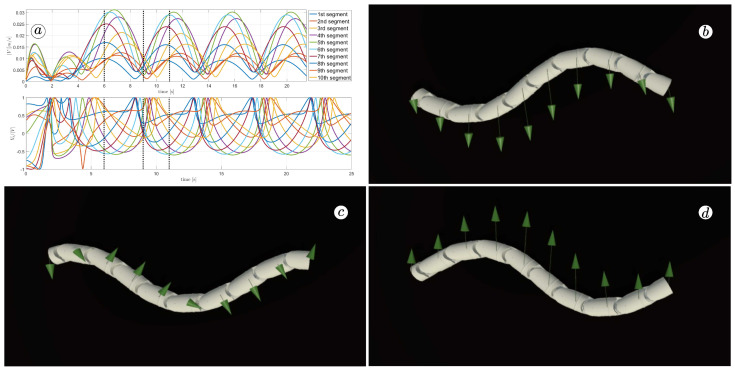
Absolute value and tangential component of segments velocities for $c_{t}=10$ and $c_{n}=10$ (**a**) and velocity vectors for t=6(s) (**b**), t=9(s) (**c**), and t=11(s) (**d**).

**Figure 20 sensors-23-03262-f020:**
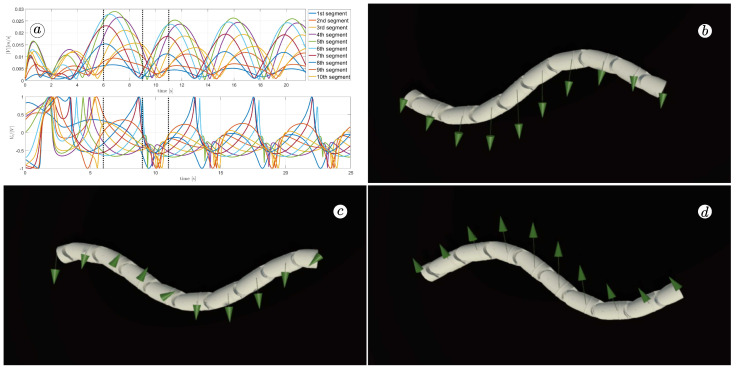
Absolute value and tangential component of segments velocities for $c_{t}=10$ and $c_{n}=1$ (**a**) and velocity vectors for t=6(s) (**b**), t=9(s) (**c**), and t=11(s) (**d**).

**Figure 21 sensors-23-03262-f021:**
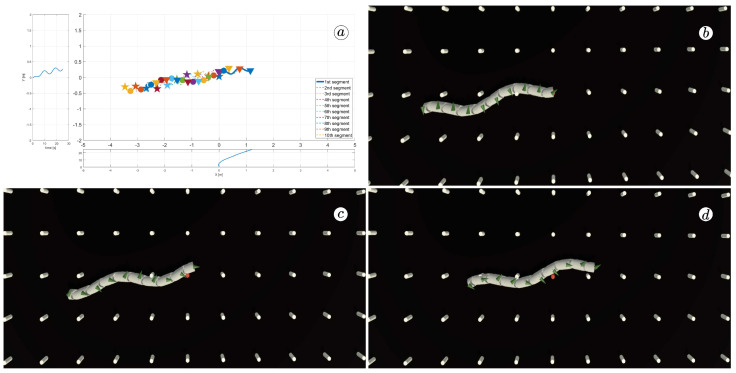
Trajectories of robot segments for $\phi_{i,\mathrm{ref}}=30^{\circ }\sin\left(40^{\circ }t+\left(i-1\right)60^{\circ }\right)$ (**a**) and robot configurations for t=5(s) (**b**), t=10(s) (**c**), and t=24(s) (**d**).

**Figure 22 sensors-23-03262-f022:**
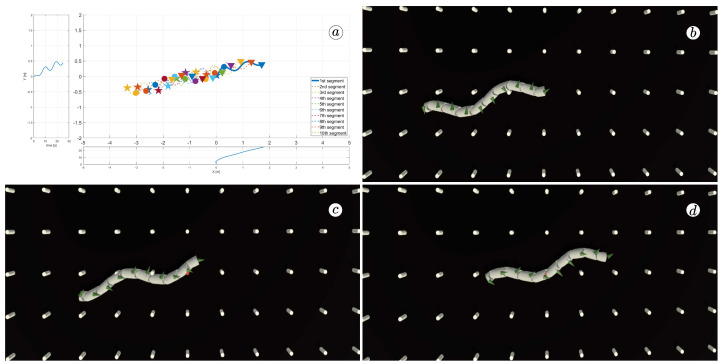
Trajectories of robot segments for $\phi_{i,\mathrm{ref}}=40^{\circ }\sin\left(40^{\circ }t+\left(i-1\right)60^{\circ }\right)$ (**a**) and robot configurations for t=5(s) (**b**), t=10(s) (**c**), and t=24(s) (**d**).

**Figure 23 sensors-23-03262-f023:**
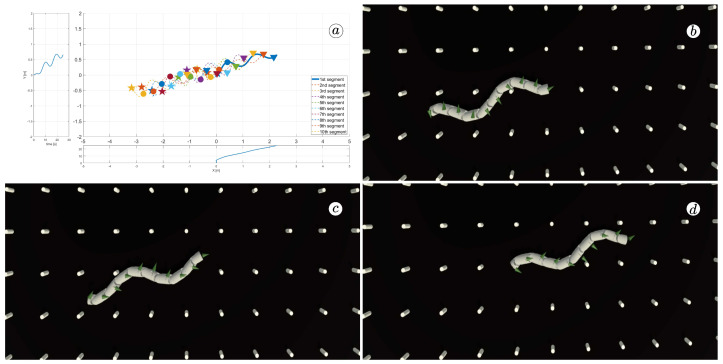
Trajectories of robot segments for $\phi_{i,\mathrm{ref}}=50^{\circ }\sin\left(40^{\circ }t+\left(i-1\right)60^{\circ }\right)$ (**a**) and robot configurations for t=5(s) (**b**), t=10(s) (**c**), and t=24(s) (**d**).

**Figure 24 sensors-23-03262-f024:**
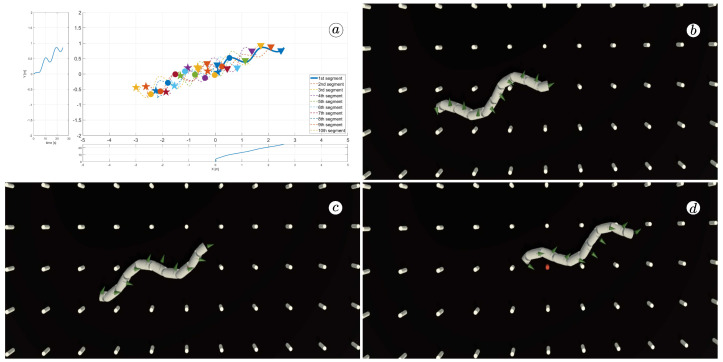
Trajectories of robot segments for $\phi_{i,\mathrm{ref}}=60^{\circ }\sin\left(40^{\circ }t+\left(i-1\right)60^{\circ }\right)$ (**a**) and robot configurations for t=5(s) (**b**), t=10(s) (**c**), and t=24(s) (**d**).

**Figure 25 sensors-23-03262-f025:**
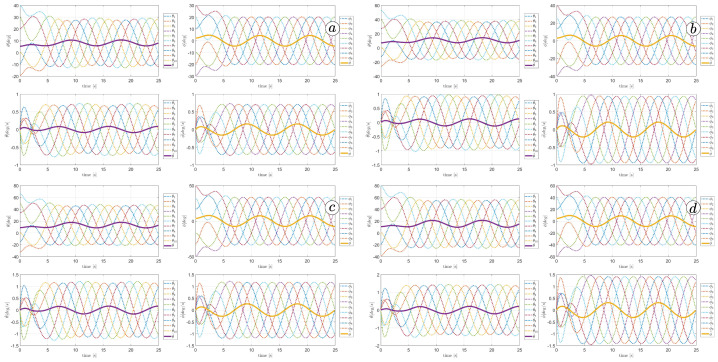
Values of link angles θ, joint angles ϕ, link angles derivatives $\dot{\theta }$, joint angles derivatives $\dot{\phi }$ for $\alpha =30^{\circ }$ (**a**), $\alpha =40^{\circ }$ (**b**), $\alpha =50^{\circ }$ (**c**), and $\alpha =60^{\circ }$ (**d**).

**Figure 26 sensors-23-03262-f026:**
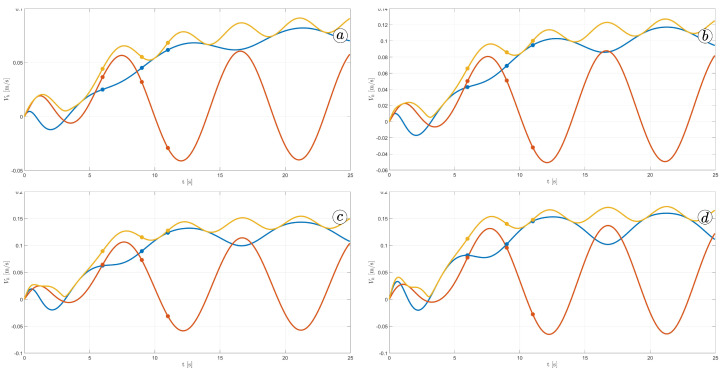
Velocity of the snake robot’s head for $\alpha =30^{\circ }$ (**a**), $\alpha =40^{\circ }$ (**b**), $\alpha =50^{\circ }$ (**c**), and $\alpha =60^{\circ }$ (**d**).

**Figure 27 sensors-23-03262-f027:**
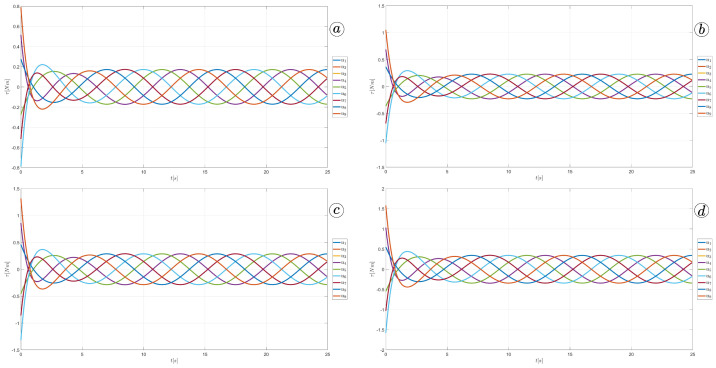
Torque in robot’s joints for $\alpha =30^{\circ }$ (**a**), $\alpha =40^{\circ }$ (**b**), $\alpha =50^{\circ }$ (**c**), and $\alpha =60^{\circ }$ (**d**).

**Figure 28 sensors-23-03262-f028:**
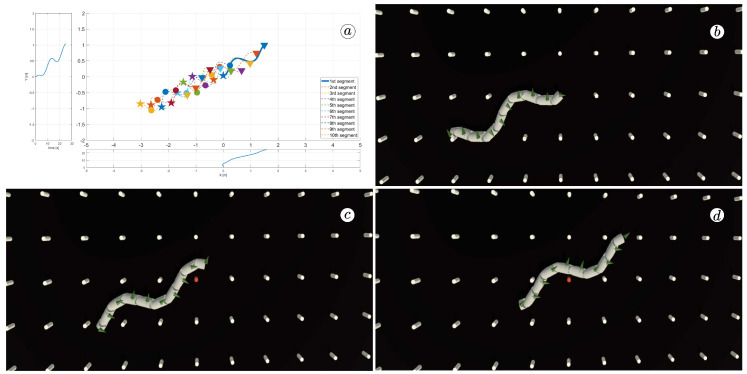
Trajectories of robot segments for $\phi_{i,\mathrm{ref}}=50^{\circ }\sin\left(30^{\circ }t+\left(i-1\right)60^{\circ }\right)$ (**a**) and robot configurations for t=5(s) (**b**), t=10(s) (**c**), and t=24(s) (**d**).

**Figure 29 sensors-23-03262-f029:**
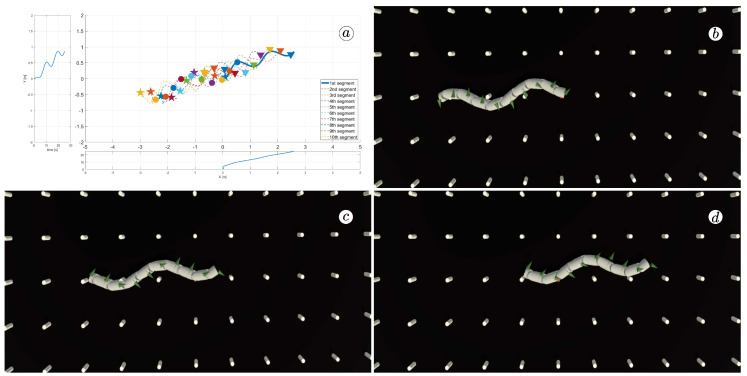
Trajectories of robot segments for $\phi_{i,\mathrm{ref}}=50^{\circ }\sin\left(50^{\circ }t+\left(i-1\right)60^{\circ }\right)$ (**a**) and robot configurations for t=5(s) (**b**), t=10(s) (**c**), and t=24(s) (**d**).

**Figure 30 sensors-23-03262-f030:**
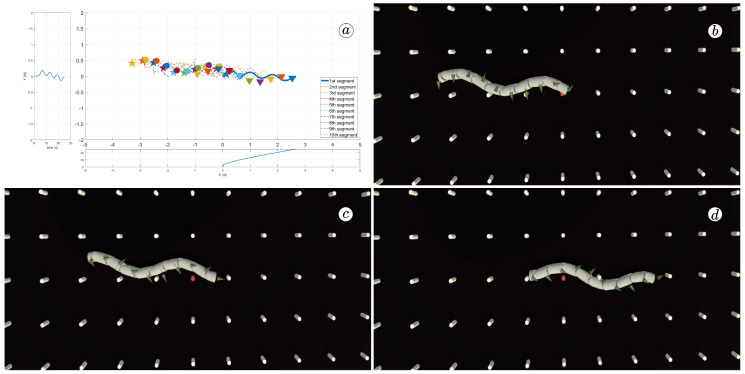
Trajectories of robot segments for $\phi_{i,\mathrm{ref}}=50^{\circ }\sin\left(60^{\circ }t+\left(i-1\right)60^{\circ }\right)$ (**a**) and robot configurations for t=5(s) (**b**), t=10(s) (**c**), and t=24(s) (**d**).

**Figure 31 sensors-23-03262-f031:**
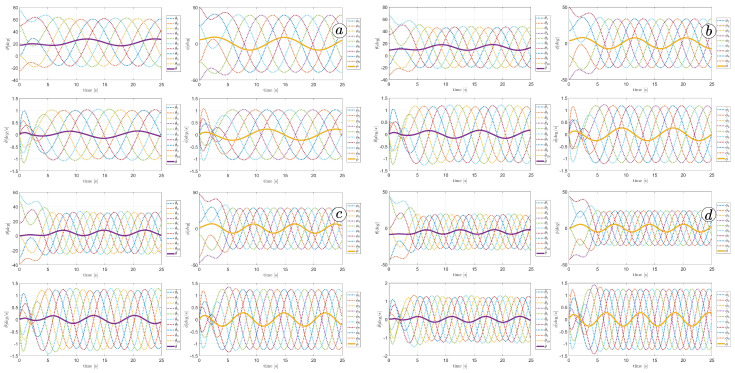
Values of link angles θ, joint angles ϕ, link angles derivatives $\dot{\theta }$, and joint angles derivatives $\dot{\phi }$ for $\omega =30^{\circ }$ (**a**), $\omega =40^{\circ }$ (**b**), $\omega =50^{\circ }$ (**c**), and $\omega =60^{\circ }$ (**d**).

**Figure 32 sensors-23-03262-f032:**
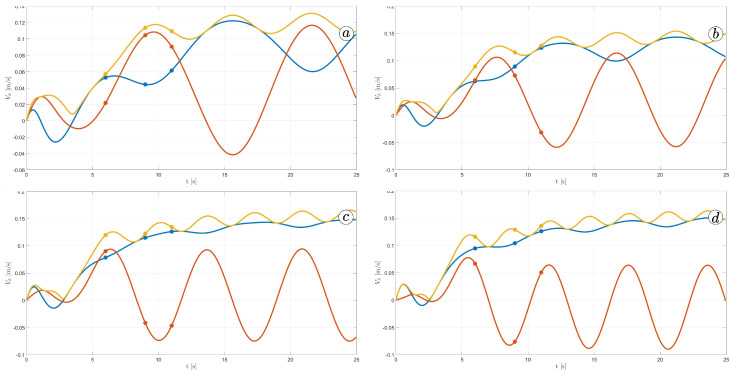
Velocity of the snake robot’s head for $\omega =30^{\circ }$ (**a**), $\omega =40^{\circ }$ (**b**), $\omega =50^{\circ }$ (**c**), and $\omega =60^{\circ }$ (**d**).

**Figure 33 sensors-23-03262-f033:**
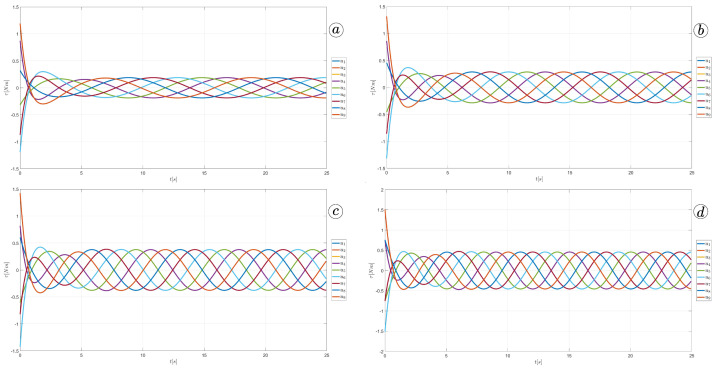
Torque in robot’s joints for $\omega =30^{\circ }$ (**a**), $\omega =40^{\circ }$ (**b**), $\omega =50^{\circ }$ (**c**), and $\omega =60^{\circ }$ (**d**).

**Figure 34 sensors-23-03262-f034:**
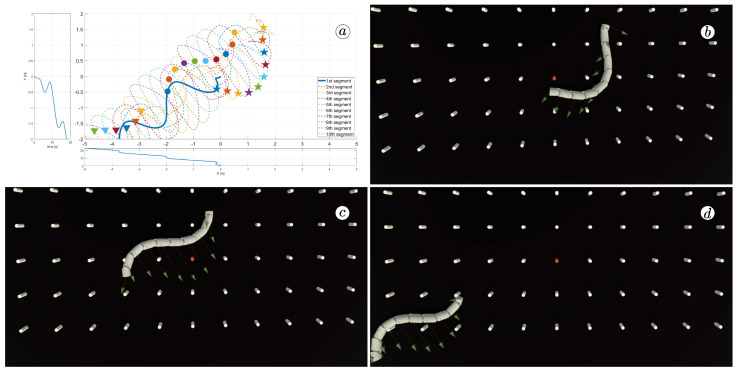
Trajectories of robot segments for $\phi_{i,\mathrm{ref}}=50^{\circ }\sin\left(50^{\circ }t+\left(i-1\right)30^{\circ }\right)$ (**a**) and robot configurations for t=5(s) (**b**), t=10(s) (**c**), and t=24(s) (**d**).

**Figure 35 sensors-23-03262-f035:**
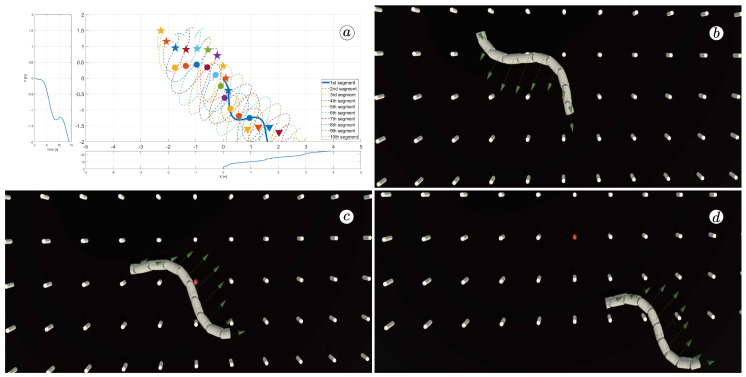
Trajectories of robot segments for $\phi_{i,\mathrm{ref}}=50^{\circ }\sin\left(50^{\circ }t+\left(i-1\right)40^{\circ }\right)$ (**a**) and robot configurations for t=5(s) (**b**), t=10(s) (**c**), and t=24(s) (**d**).

**Figure 36 sensors-23-03262-f036:**
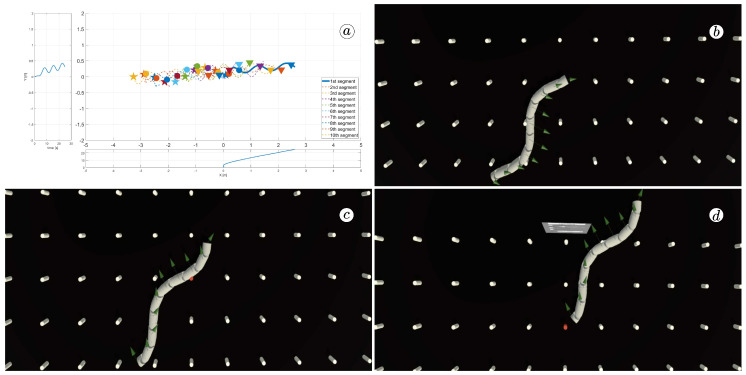
Trajectories of robot segments for $\phi_{i,\mathrm{ref}}=50^{\circ }\sin\left(50^{\circ }t+\left(i-1\right)50^{\circ }\right)$ (**a**) and robot configurations for t=5(s) (**b**), t=10(s) (**c**), and t=24(s) (**d**).

**Figure 37 sensors-23-03262-f037:**
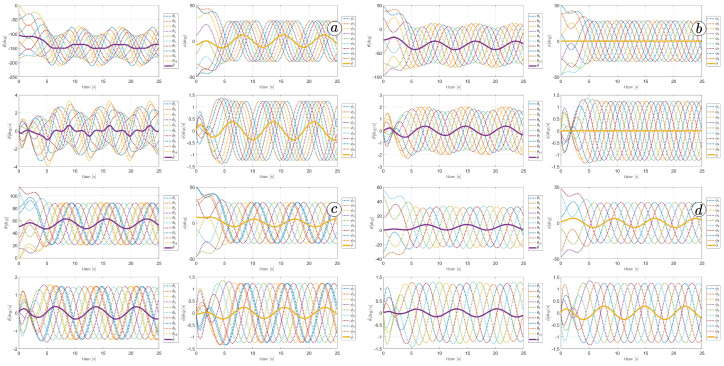
Values of link angles θ, joint angles ϕ, link angles derivatives $\dot{\theta }$, and joint angles derivatives $\dot{\phi }$ for $\delta =30^{\circ }$ (**a**), $\delta =40^{\circ }$ (**b**), $\delta =50^{\circ }$ (**c**), and $\delta =60^{\circ }$ (**d**).

**Figure 38 sensors-23-03262-f038:**
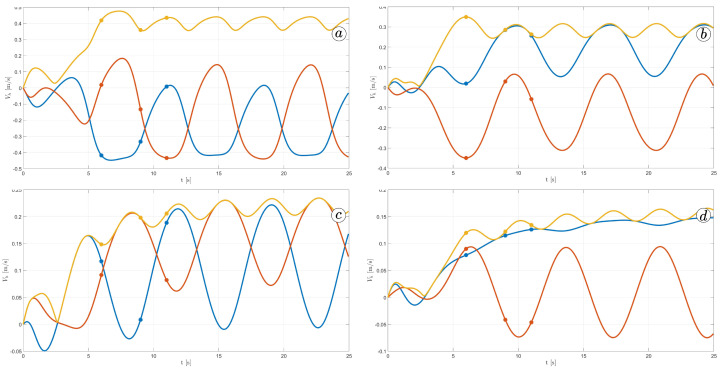
Velocity of the snake robot’s head for $\delta =30^{\circ }$ (**a**), $\delta =40^{\circ }$ (**b**), $\delta =50^{\circ }$ (**c**), and $\delta =60^{\circ }$ (**d**).

**Figure 39 sensors-23-03262-f039:**
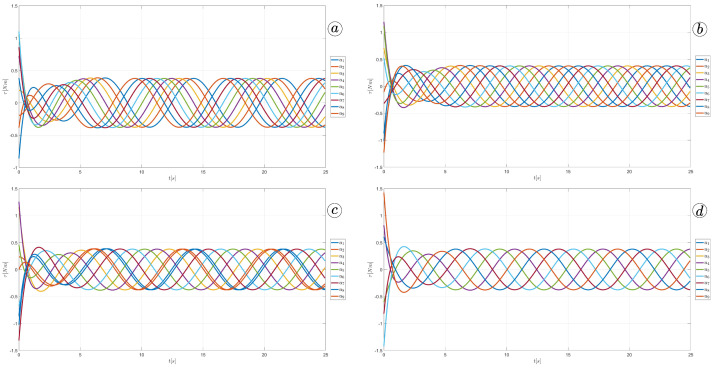
Torque in robot’s joints for $\delta =30^{\circ }$ (**a**), $\delta =40^{\circ }$ (**b**), $\delta =50^{\circ }$ (**c**), and $\delta =60^{\circ }$ (**d**).

**Figure 40 sensors-23-03262-f040:**
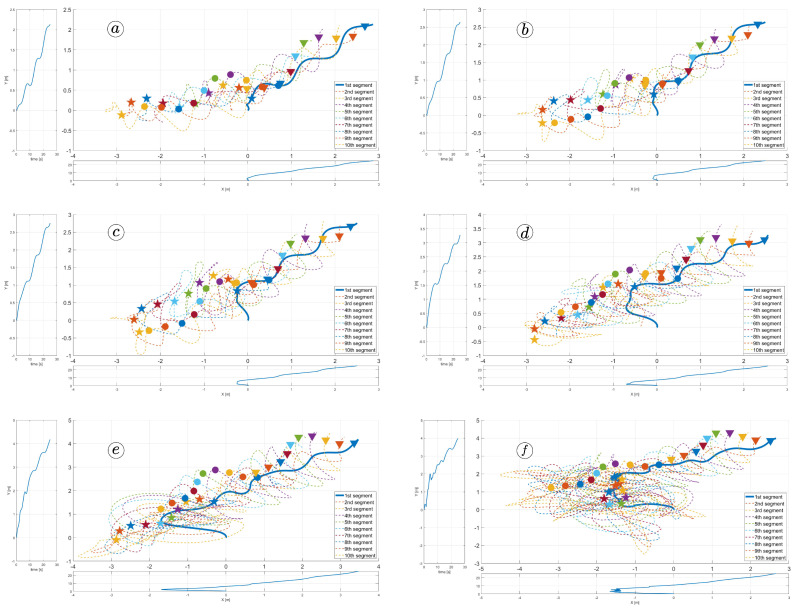
Trajectories of robot segments for $k_{\theta }=0.1$ (**a**), $k_{\theta }=0.3$ (**b**), $k_{\theta }=0.5$ (**c**), $k_{\theta }=1$ (**d**), $k_{\theta }=2$ (**e**), and $k_{\theta }=3$ (**f**).

**Figure 41 sensors-23-03262-f041:**
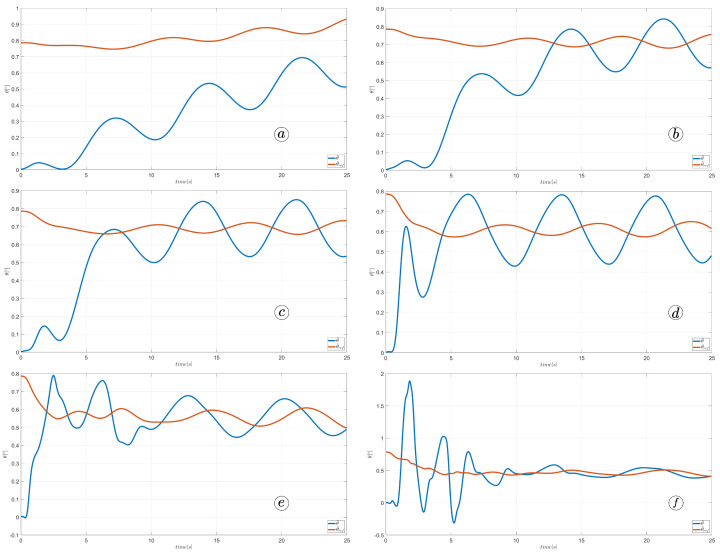
Heading angle and reference heading angle for $k_{\theta }=0.1$ (**a**), $k_{\theta }=0.3$ (**b**), $k_{\theta }=0.5$ (**c**), $k_{\theta }=1$ (**d**), $k_{\theta }=2$ (**e**), and $k_{\theta }=3$ (**f**).

**Figure 42 sensors-23-03262-f042:**
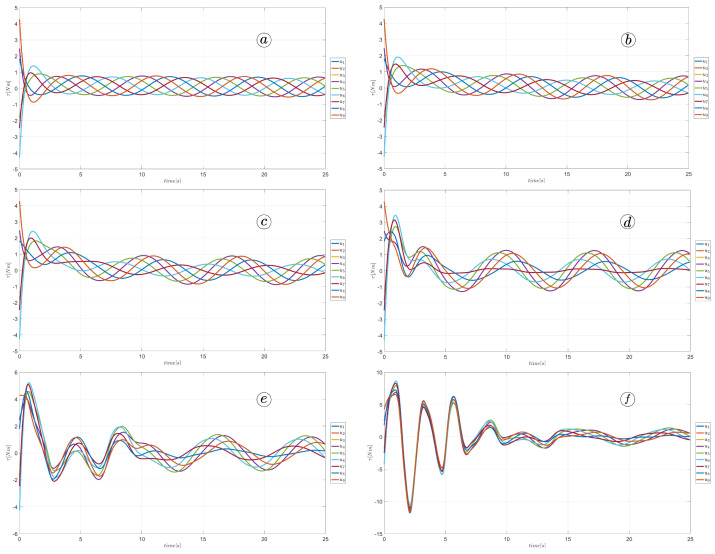
Torque in robot’s joints for $k_{\theta }=0.1$ (**a**), $k_{\theta }=0.3$ (**b**), $k_{\theta }=0.5$ (**c**), $k_{\theta }=1$ (**d**), $k_{\theta }=2$ (**e**), and $k_{\theta }=3$ (**f**).

**Figure 43 sensors-23-03262-f043:**
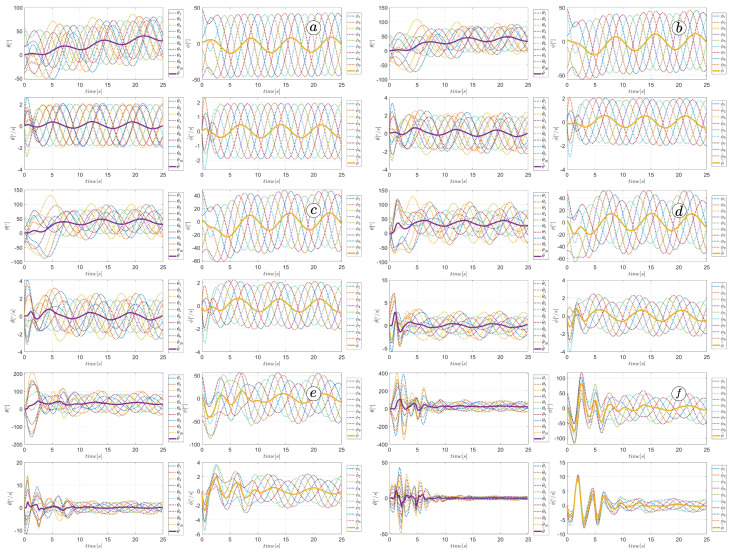
Values of link angles θ, joint angles ϕ, link angles derivatives $\dot{\theta }$, joint angles derivatives $\dot{\phi }$ for $k_{\theta }=0.1$ (**a**), $k_{\theta }=0.3$ (**b**), $k_{\theta }=0.5$ (**c**), $k_{\theta }=1$ (**d**), $k_{\theta }=2$ (**e**), and $k_{\theta }=3$ (**f**).

**Table 1 sensors-23-03262-t001:** Simulation results for different amplitudes α.

α	ω	δ	*d*	max(|u|)	*V*	max(|θ|)	$\max(|\dot{\theta }|)$	max(|ϕ|)	$\max(|\dot{\phi }|)$
$30^{\circ }$	$40^{\circ }$	$60^{\circ }$	1.25	0.17	0.091	$29^{\circ }$	$0.74^{\circ }/s$	$20^{\circ }$	$0.7^{\circ }/s$
$40^{\circ }$	$40^{\circ }$	$60^{\circ }$	1.86	0.23	0.125	$39^{\circ }$	$0.98^{\circ }/s$	$27^{\circ }$	$0.94^{\circ }/s$
$50^{\circ }$	$40^{\circ }$	$60^{\circ }$	2.36	0.29	0.15	$48^{\circ }$	$1.21^{\circ }/s$	$34^{\circ }$	$1.17^{\circ }/s$
$60^{\circ }$	$40^{\circ }$	$60^{\circ }$	2.73	0.34	0.16	$57^{\circ }$	$1.44^{\circ }/s$	$40^{\circ }$	$1.4^{\circ }/s$

**Table 2 sensors-23-03262-t002:** Simulation results for different angular frequencies ω.

α	ω	δ	*d*	max(|u|)	*V*	max(|θ|)	$\max(|\dot{\theta }|)$	max(|ϕ|)	$\max(|\dot{\phi }|)$
$50^{\circ }$	$30^{\circ }$	$60^{\circ }$	1.9	0.19	0.11	$62^{\circ }$	$1.06^{\circ }/s$	$39^{\circ }$	$1.03^{\circ }/s$
$50^{\circ }$	$40^{\circ }$	$60^{\circ }$	2.36	0.29	0.15	$48^{\circ }$	$1.21^{\circ }/s$	$34^{\circ }$	$1.17^{\circ }/s$
$50^{\circ }$	$50^{\circ }$	$60^{\circ }$	2.62	0.38	0.16	$33^{\circ }$	$1.28^{\circ }/s$	$28^{\circ }$	$1.24^{\circ }/s$
$50^{\circ }$	$60^{\circ }$	$60^{\circ }$	2.67	0.46	0.15	$30^{\circ }$	$1.33^{\circ }/s$	$24^{\circ }$	$1.25^{\circ }/s$

**Table 3 sensors-23-03262-t003:** Simulation results for different phase shifts δ.

α	ω	δ	*d*	max(|u|)	*V*	max(|θ|)	$\max(|\dot{\theta }|)$	max(|ϕ|)	$\max(|\dot{\phi }|)$
$50^{\circ }$	$50^{\circ }$	$30^{\circ }$	6.37	0.38	0.43	$212^{\circ }$	$3.33^{\circ }/s$	$28^{\circ }$	$1.24^{\circ }/s$
$50^{\circ }$	$50^{\circ }$	$40^{\circ }$	4.99	0.38	0.29	$80^{\circ }$	$2.02^{\circ }/s$	$28^{\circ }$	$1.24^{\circ }/s$
$50^{\circ }$	$50^{\circ }$	$50^{\circ }$	3.77	0.38	0.21	$89^{\circ }$	$1.48^{\circ }/s$	$28^{\circ }$	$1.24^{\circ }/s$
$50^{\circ }$	$50^{\circ }$	$60^{\circ }$	2.63	0.4	0.16	$33^{\circ }$	$1.28^{\circ }/s$	$28^{\circ }$	$1.24^{\circ }/s$

**Table 4 sensors-23-03262-t004:** Simulation results for different control gains.

$k_{p}$	$k_{d}$	*d*	max(|u|)	*V*	max(|θ|)	$\max(|\dot{\theta }|)$	max(|ϕ|)	$\max(|\dot{\phi }|)$
0.5	0.5	3.05	0.67	0.22	$62^{\circ }$	$2.33^{\circ }/s$	$54^{\circ }$	$2.227^{\circ }/s$
1	0.5	0.9	1.36	0.13	$114^{\circ }$	$4.56^{\circ }/s$	$101^{\circ }$	$4.41^{\circ }/s$
3	0.5	3.27	1.20	0.18	$121^{\circ }$	$6.55^{\circ }/s$	$70^{\circ }$	$3.24^{\circ }/s$
5	0.5	3.72	1.11	0.19	$110^{\circ }$	$8.9^{\circ }/s$	$61^{\circ }$	$3.06^{\circ }/s$
0.5	1	2.49	0.37	0.16	$56^{\circ }$	$1.35^{\circ }/s$	$28^{\circ }$	$1.22^{\circ }/s$
1	1	3.2	0.74	0.22	$63^{\circ }$	$2.45^{\circ }/s$	$55^{\circ }$	$2.42^{\circ }/s$
3	1	3.33	0.85	0.19	$88^{\circ }$	$4.47^{\circ }/s$	$63^{\circ }$	$2.74^{\circ }/s$
5	1	3.59	0.78	0.19	$86^{\circ }$	$6.38^{\circ }/s$	$58^{\circ }$	$2.55^{\circ }/s$
0.5	3	0.82	0.13	0.04	$56^{\circ }$	$0.68^{\circ }/s$	$18^{\circ }$	$0.27^{\circ }/s$
1	3	1.79	0.25	0.1	$56^{\circ }$	$1.10^{\circ }/s$	$20^{\circ }$	$0.84^{\circ }/s$
3	3	3.43	0.58	0.2	$56^{\circ }$	$2.71^{\circ }/s$	$44^{\circ }$	$1.90^{\circ }/s$
5	3	3.57	0.67	0.19	$58^{\circ }$	$4.14^{\circ }/s$	$50^{\circ }$	$2.19^{\circ }/s$
0.5	5	0.5	0.08	0.02	$56^{\circ }$	$0.46^{\circ }/s$	$24^{\circ }$	$0.33^{\circ }/s$
1	5	1	0.15	0.05	$56^{\circ }$	$0.78^{\circ }/s$	$18^{\circ }$	$0.55^{\circ }/s$
3	5	2.85	0.41	0.17	$56^{\circ }$	$1.97^{\circ }/s$	$31^{\circ }$	$1.34^{\circ }/s$
5	5	3.4	0.55	0.19	$56^{\circ }$	$3.1^{\circ }/s$	$41^{\circ }$	$1.80^{\circ }/s$

## Data Availability

Not applicable.

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
