# Peer review of "Analysis of the Snake Robot Kinematics with Virtual Reality Visualisation"

_sensors, 2023, doi:10.3390/s23063262_

Round 1

Reviewer 1 Report

Well crafted paper, can see no needed improvements

Author Response

The authors express their heartfelt gratitude to the reviewers for reviewing the paper and positively assessing its content.

Reviewer 2 Report

The authors performed a kinematic simulation of the motion of a snake robot on a flat surface in MATLAB to study the impact of model parameters on the angular and linear velocities of robot segments. The authors concluded that VR-based simulation shall improve the control algorithms for robotic systems.

Articles 19 to 26 are placed in the introduction without detailed discussions; please elaborate and include some recent literature on AR/VR for manufacturing applications.

Makhataeva, Zhanat, and Huseyin Atakan Varol. "Augmented reality for robotics: A review." Robotics 9.2 (2020): 21.

Eswaran, M., and MVA Raju Bahubalendruni. "Challenges and opportunities on AR/VR technologies for manufacturing systems in the context of industry 4.0: A state of the art review." Journal of Manufacturing Systems 65 (2022): 260-278.

Sonawani, Shubham, and Heni Amor. "When And Where Are You Going? A Mixed-Reality Framework for Human-Robot Collaboration." 5th International Workshop on Virtual, Augmented, and Mixed Reality for HRI. 2022.

Give the illustrations to represent the snake robot with joint & link angles  & friction forces.

It is difficult to follow the equations without proper graphical illustrations.

Give the pseudocode/procedure for simulation.

The results presented in figure 7 to 10 are interesting,

Please provide the sub-captions for the figures like (a), (b), (c)..

Please do a comparative assessment with similar prior art to draw the merits of the proposed work

Author Response

The authors sincerely thank the reviewers for their invaluable comments and insightful suggestions. We have made every effort to address all questions raised by the reviewers, and most of the amendments have been indicated in the revised text for ease of reference. The manuscript has been substantially enhanced due to the thoughtful feedback received, and all issues raised by the reviewers have been addressed in the attached PDF file. Once again, we express our sincere thanks for your time and expertise in reviewing our work.

Reviewer 3 Report

The manuscript discusses an interesting topic. The manuscript is organised and evaluated using a scientific approach. 

Changes:

1. Virtual Reality (VR) is abbreviated many times in manuscript (lines no 18, 30, 44, 51 and 56).

2. The introduction discusses a lot of application information. It is better to make it crisp and short. 

3. I suggest conducting some comparative study if possible.

4. The conclusion section should be extended with future work direction. 

Author Response

(The authors gave the same response as above.)

Reviewer 4 Report

Authors present the outcomes of engineering simulations that focuses on the motion of a snake robot on a flat surface. The simulation results using Matlab software provide a detailed analysis of the kinematics of the snake robot and reveal the impact of model parameters on the angular and linear velocities of robot segments. The VR simulation offers a more immersive experience, allowing the viewer to observe the simulation results and modify the simulation parameters within the VR environment.

This is a good paper, however, I have some comments and remarks.

- Abstract needs to be rewritten to understand the summary of the work done highlighting the principal contributions. 

- The Snake robot model as well as the used controller are known. Authors should clarify the contribution to this subject. 

- The graphical user interface is good, but it need to be discussed further. 

- English is generally good, but needs to be polished further. The manuscript should be formatted better and some spelling and grammar should be checked carefully.

- Discussion part is too reduced, this can be extended by adding more details. 

- The Conclusion should be rewritten by integrating the limitations and the perspective. 

Concluding, the paper has potential to be appreciated by the readers and the above comments are formulated such that to enhance its impact.

Author Response

(The authors gave the same response as above.)

Round 2

Reviewer 2 Report

The manuscript is well revised, and the responses were satisfactory and no further comments to the authors.

All the best!.